# SJD-SV: Speculative Jacobi Decoding with Semantics Verification for Autoregressive Image Generation

**Baoquan Zhang** [1] **Bingqi Shan** [1 2] **Shihao Fang** [1] **Kenghong Lin** [1] **Xutao Li** [1] **Yunming Ye** [1]

## Abstract

Speculative Jacobi Decoding (SJD) is an important approach for accelerating autoregressive image generation. Although SJD has shown superior performance, recent studies point out that it usually suffers from a token ambiguity issue during token verification but its reason can not be well explained. To figure out this reason, in this paper, we conduct a visualization analysis on vision token and find that different from text tokens, vision tokens generally corresponds to some local, small, and unclear vision details, which means only using single token is difficult to accurately express a certain semantic, thereby causing token ambiguity issue. To this end, we propose a novel Speculative Jacobi Decoding with Semantics Verification (called SJD-SV), for accelerating autoregressive image generation. The key idea is that leveraging the strong correction characters between tokens to recognize semantic-aware token subsequence and then instead of perform token-by-token verification, turning to perform verification on semantic-aware token subsequence level for accelerating image generation. In particular, our method is plug-in, which can be directly integrated into existing SJD and its variants. Extensive experiments on various datasets show that existing SJD methods achieve significant performance improvement after integrating our SJD-SV method.

## 1. Introduction

Autoregressive (AR) models have recently emerged as a powerful paradigm for image generation (Esser et al., 2021), achieving strong performance in high-fidelity synthesis and

[1]Shenzhen Key Laboratory of Internet Information Collaboration, Harbin Institute of Technology, Shenzhen [2]Pengcheng Laboratory, Shenzhen. Correspondence to: Kenghong Lin <linkenghong@stu.hit.edu.cn>, Xutao Li <lixutao@hit.edu.cn>.

*Proceedings of the 43rd International Conference on Machine Learning*, Seoul, South Korea. PMLR 306, 2026. Copyright 2026 by the author(s).

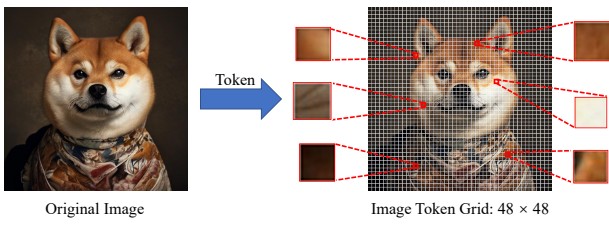

(a) Visualization Analysis on Vision Tokens

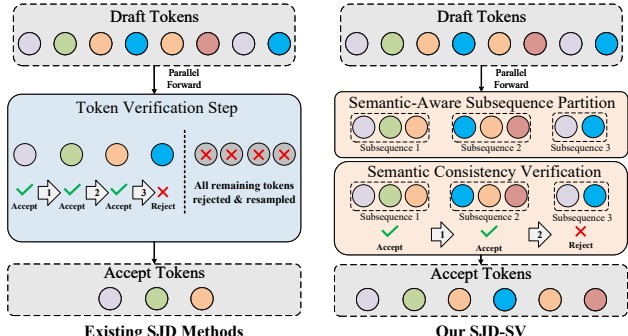

(b) Existing SJD Method vs Our SJD-SV

*Figure 1.* (a) Existing image AR models represents an image as a fine-grained token sequence, such that individual tokens is difficult to accurately and clearly express a certain semantic, thereby introducing token ambiguity; (b) Instead of performing token-by-token verification as existing SJD methods, we turn to perform verification on semantic-aware token subsequence level. Its advantage is that token subsequences can encode more stable semantic information, effectively reducing false rejections during verification.

complex semantic modeling (Sun et al., 2024b; Yu et al., 2024; Team, 2024). However, despite their modeling advantages, AR image generation remains fundamentally limited by slow inference, as images are generated token by token in a long sequential process (Van Den Oord et al., 2016; Chang et al., 2022). To mitigate this issue, Speculative Jacobi Decoding (SJD) (Leviathan et al., 2023; Teng et al., 2024b) has been proposed as a promising acceleration strategy, which drafts multiple future tokens in parallel and verifies them sequentially. While SJD can substantially reduce decoding latency, its practical speedup is often constrained by a low token acceptance rate during verification, leaving significant acceleration potential (Jang et al., 2024b; So et al., 2025).

Recently, existing studies have pointed out that the low token acceptance rate mainly stems from token ambiguity, i.e.,

visual autoregressive models tend to assign low confidence to individual tokens, with multiple tokens being plausible candidates at each step. To mitigate this issue, these works attempt to improve acceptance rates by relaxing the verification criteria. For example, Jiang et al. (Jang et al., 2024b) propose a latent neighbor-based relaxation strategy, which accepts drafted tokens that fall within a tolerance region of the verifier's latent space, alleviating rejections caused by minor token-level discrepancies. So et al. (So et al., 2025) introduce Group-based relaxation strategy, which performs verification over token groups rather than individual tokens to relax strict constraints. Although these SJD variants have achieve superior performance, the underlying reason causing the above token ambiguity remain is unclear.

To figure out this reason, in this paper, as shown in Figure 1(a), we conduct a visualization analysis on vision tokens and find that 1) existing image AR models (e.g., Lumina-mGPT) generally represents an image as a very long token sequence (e.g., $48 \times 48$ token-grid sequence), which is because only such fine-grained spatial tokens can faithfully capture the high-frequency visual details and local structural variations present in natural images; and 2) however, such a fine-grained visual token grid causes that each token corresponds to only a very small local image patch, which is difficult to accurately express a certain semantic. As a result, individual tokens are weakly constrained by semantic context, making multiple tokens perceptually and semantically interchangeable for the same spatial location, thereby introducing the token ambiguity issue.

Based on the above fact, we present a novel Speculative Jacobi Decoding with Semantics Verification (called SJD-SV), for accelerating autoregressive image generation. The key idea is that instead of performing token-by-token verification as existing SJD and its variants, as shown in Figure 1(b), we turn to perform verification on semantic-aware token subsequence level. Specifically, we first partitions the drafted token sequence into a set of semantic-aware token subsequences, where each subsequence corresponds to a contiguous spatial region and jointly represents a coherent local semantic unit. Then, during speculative decoding, instead of verifying each drafted token independently, we evaluates semantic consistency at the above token subsequence level for accelerating autoregressive image generation by assessing the joint likelihood of token subsequences with respect to the target model. The advantage of such design is that although individual tokens are ambiguous and interchangeable, a short token subsequence jointly encodes more stable and coherent semantic information, such that semantic consistency can be reliably assessed even in the presence of token-level variations, effectively reducing false rejections during speculative verification.

Our main contributions can be summarized as follows:

- We reveal the root cause of token ambiguity in visual AR models and demonstrate that semantic-aware verification, rather than token-level matching, is the critical factor for accurate and efficient speculative decoding.

- Instead of performing token-level verification, we shift verification to the token-subsequence level. This allows semantic consistency to be checked over token subsequences, effectively mitigating token ambiguity and improving speculative decoding efficiency.

- We conduct comprehensive experiments on multiple benchmark datasets and shows that our method achieves state-of-the-art (SOTA) performance.

**Conflict of Interest Disclosure:** The authors declare that they have no relevant financial or non-financial conflicts of interest to disclose.

## 2. Related Work

### 2.1. Autoregressive Image Generation

Autoregressive image generation formulates visual synthesis as sequential token prediction. This paradigm leverages the powerful sequence modeling capabilities of Transformers and has emerged as a promising direction for controllable image synthesis (Chen et al., 2020; Ramesh et al., 2021; Yu et al., 2021) . Existing autoregressive image generation methods can be roughly divided into two groups: **1) Continuous-token autoregressive models.** These approaches directly model the distribution over continuous pixel values or latent representations without discrete quantization. Early pioneering works such as PixelSNAIL (Chen et al., 2018) and ImageGPT (Chen et al., 2020) demonstrate that autoregressive modeling in continuous space can capture complex visual dependencies. More recently, MAR (Li et al., 2024) explores masked autoregressive generation in diffusion latent space, achieving competitive performance with diffusion models. **2) Discrete-token autoregressive models.** This line of works first encodes images into discrete token sequences via learned codebooks, then applies autoregressive Transformers for generation. LlamaGen (Sun et al., 2024b) scales up vanilla autoregressive Transformers with next-token prediction, demonstrating strong scaling properties. VAR (Tian et al., 2024) introduces a coarse-to-fine "next-scale prediction" paradigm that generates tokens at progressively finer resolutions. Unified multimodal models such as Emu3 (Wang et al., 2024) and Show-o (Xie et al., 2024) further extend discrete autoregressive generation by jointly modeling text and image tokens within a single vocabulary. Despite these advances in generation quality and controllability, the sequential decoding process remains a fundamental efficiency bottleneck, particularly for high-resolution synthesis. In this paper, we focus on

accelerating this autoregressive image decoding process.

## 2.2. Autoregressive Generation Acceleration

Autoregressive generation acceleration seeks to mitigate the inference latency inherent in token-by-token decoding, which scales linearly with sequence length and becomes prohibitive for dense visual token grids. Existing acceleration approaches can be roughly divided into two groups: **1) Model-based acceleration methods.** These approaches modify model architectures or training objectives to enable faster generation. For instance, MAR (Li et al., 2024) employs a masked prediction strategy that allows parallel token generation within each step. HART (Tang et al., 2025) introduces a hybrid tokenizer that decomposes images into discrete semantic tokens and continuous residual tokens, reducing the sequence length for autoregressive modeling. In addition, a series of methods accelerate generation through dedicated model designs for parallel prediction, including iterative masked decoding in MaskGIT (Chang et al., 2022) and MAGVIT (Yu et al., 2023), parallel execution of multiple prediction operations in PAR (Wang et al., 2025) and LPD (Zhang et al., 2025), and flexible generation orders together with parallel decoding strategies in RandAR (Pang et al., 2025) and ARPG (Li et al., 2025). **2) Inference-based acceleration methods.** These approaches accelerate generation at inference time without architectural modifications or retraining. Speculative decoding (Teng et al., 2024a), originally developed for language models, has been adapted to visual autoregressive generation. This paradigm uses a lightweight draft model to propose candidate tokens in parallel, which are then verified by the target model. Speculative Jacobi Decoding (Teng et al., 2024a) reformulates autoregressive generation as a fixed-point iteration problem, enabling parallel token updates via only a single model. Although SJD achieves significant acceleration performance, recent advancements like GSD (So et al., 2025) and LANTERN (Jang et al., 2024b) argue that its verification standard is overly stringent. This strictness leads to high rejection rates, as the model often rejects feasible tokens. To address this limitation, these variants propose enhancing SJD's efficiency by relaxing the verification criteria, thereby allowing for higher acceptance rates.

However, existing relaxation methods mitigate rejections but overlook the root cause: the inherent semantic ambiguity of fine-grained tokens. To address this, we propose SJD-SV, which shifts verification from isolated tokens to semantic-aware token subsequences. Crucially, our approach addresses the ambiguity issue from the perspective of verification granularity. As this optimization is fundamentally independent of the verification strictness targeted by relaxation strategies, SJD-SV serves as a flexible plug-in module that can be seamlessly integrated into existing SJD variants to yield substantial further acceleration gains.

## 3. Preliminaries

### 3.1. Problem Formulation

This work addresses the challenge of accelerating autoregressive image generation without compromising output quality. Let $x$ denote a conditioning input, and let $y = (y_1, y_2, \ldots, y_N)$ represent the target image token sequence. An autoregressive model factorizes the joint distribution as:

$$p(y \mid c) = \prod_{n=1}^{N} p(y_n \mid y_1, \ldots, y_{n-1}, x). \quad (1)$$

This formulation necessitates $N$ sequential forward passes to generate the complete sequence, resulting in substantial inference latency that scales linearly with the token count. For high-resolution image synthesis where $N$ can reach 1024 or more, this sequential bottleneck becomes prohibitively expensive. In this paper, we operate under the *training-free* constraint, where the pretrained autoregressive model is frozen and no fine-tuning or architectural changes are permitted. Under this setting, SJD has demonstrated promising acceleration capabilities. We provide a brief overview of SJD in the following subsection.

### 3.2. Speculative Jacobi Decoding

Speculative Jacobi Decoding (SJD) (Teng et al., 2024a) is a training-free method that accelerates autoregressive image generation by enabling parallel token updates within a single model. The core idea of SJD follows a two-phase "draft-then-verify" procedure: given a current prefix sequence, SJD first speculatively predicts a winodw of $K$ tokens during the drafting phase, and then employs the same autoregressive model to sequentially verify whether each drafted token matches the model's own prediction. Tokens that pass verification are appended to the accepted prefix, while the first mismatched token triggers rejection of all subsequent tokens in the block. The process then iterates from the updated prefix until the entire sequence is generated.

While SJD offers notable speedups over vanilla autoregressive decoding, its acceleration is fundamentally bounded by the token acceptance rate during verification. Empirically, existing studies have observed that visual autoregressive models exhibit low acceptance rates compared to their language counterparts, which significantly undermines the efficiency gains of SJD. Recent investigations attribute this phenomenon to *token ambiguity* (Jang et al., 2024b; So et al., 2025). Specifically, this refers to a situation where multiple candidate tokens exhibit comparable probabilities and appear equally plausible for a given position. However, the underlying mechanism that causes such ambiguity in visual tokenization remains poorly understood. This raises a critical question: *What is the root cause of token ambiguity in visual autoregressive models, and how can we exploit*

*this understanding to improve token acceptance rates and achieve more effective acceleration?*

## 4. The Proposed SJD-SV Method

As analyzed in Section 1, to achieve high-quality image generation, existing image AR models (e.g., Lumina-mGPT) typically represent an image as a very long token sequence (e.g., a $48 \times 48$ token grid, see Figure 1(a)). Such fine-grained tokens imply that each visual token corresponds to only a small local image patch, which is often insufficient to accurately and unambiguously express a specific semantic concept. Consequently, individual tokens tend to be semantically under-specified, leading to inherent token ambiguity. To address this issue, we propose Speculative Jacobi Decoding with Semantic Verification (SJD-SV), a novel approach that shifts speculative verification from individual tokens to semantic-aware token subsequences. The key advantage of this design is that the token subsequence can jointly encode more stable and coherent semantic information, such that semantic consistency to be reliably assessed even in the presence of token-level variations, thereby effectively reducing false rejections during speculative verification.

Specifically, as shown in **Algorithm 1**, during each Jacobi iteration, SJD-SV proceeds in two steps. First, given a drafted token sequence $\hat{x}$, our SJD-SV partitions it into a set of semantic-aware token subsequences $\mathcal{S}$, where each subsequence $\mathcal{S}_k$ corresponds to a contiguous spatial region and jointly encodes a coherent local semantic unit. Second, during speculative verification, SJD-SV no longer verifies drafted tokens individually. Instead, it evaluates semantic consistency at the token-subsequence level by assessing the joint likelihood of each subsequence with respect to the target model. By verifying subsequences rather than individual tokens, our SJD-SV tolerates minor token-level variations while preserving semantic correctness, thereby enabling more efficient autoregressive image generation. Next, we elaborate on these two key steps in detail.

### 4.1. Semantic-Aware Subsequence Partition

The central challenge of our SJD-SV method lies in how to partition a drafted token sequence into meaningful semantic subsequence. Intuitively, within a coherent semantic unit, as earlier tokens are generated and confirmed, the uncertainty of subsequent tokens decreases, i.e., token prediction probability increases along the subsequence. For example, when generating a common phrase such as "the capital of France is", once the model has produced the tokens "the capital of France is", the subsequent token "Paris" becomes highly predictable and is assigned a substantially higher probability than other candidates. Thus, the continuous upward trend in token probabilities can be regarded as a key feature for partitioning token sequences into meaningful semantic

---

**Algorithm 1** SJD-SV

**Input:** Drafted tokens $\hat{x}$, Target/Draft distributions $q, p$, Threshold $\alpha$
**Output:** Final verified sequence $x_{out}$

1: *// Step 1: Semantic-Aware Subsequence Partition*
2: $\mathcal{S} \leftarrow$ **Partition**$(\hat{x}, q)$
3: $x_{out} \leftarrow \emptyset$
4: *// Step 2: Semantic Verification on Token Subsequence*
5: **for** $S_k \in \mathcal{S}$ **do**
6:     Let $S_k$ be $\hat{x}_{b:e}$
7:     *// Verify unit with fallback (Call Alg. 2)*
8:     $e' \leftarrow$ **FallbackVerify**$(S_k, \alpha)$
9:
10:     Append $\hat{x}_{b:e'}$ to $X_{out}$
11:     **if** $e' < e$ **then**
12:         Sample $x_{new} \sim \text{norm}(\max(0, q_{e'+1} - p_{e'+1}))$
13:         Append $x_{new}$ to $x_{out}$
14:         **break**
15:     **end if**
16: **end for**
17: **return** $x_{out}$

---

subsequences. To verify this point, we randomly sample a text prompt from the Parti-prompt dataset and visualize the corresponding original image patches for several token subsequences partitioned using the continuous upward trend of token probabilities, along with their individual tokens. As shown in Figure 2(a), token subsequences identified using this probability-increasing feature correspond to more semantically coherent units compared to individual tokens.

Based on the above idea, we design a token probability-based subsequence partitioning strategy to identify semantic-aware token subsequences. Formally, let $\hat{x}_{1:L}$ denote a drafted token sequence generated by the Jacobi drafting step. During the verification step, the model computes the probability distribution $q_i$ for each token position $i$, conditioned on the preceding drafted tokens. Using this token probability $q_i(\hat{x}_i)$ at sampled token $\hat{x}_i$, we partition the drafted token sequence $\hat{x}_{1:L}$ by grouping its consecutive tokens that exhibit a non-decreasing probability trend. Specifically, starting from a position $b$, we extend the subsequence until a drop in confidence is encountered. The resulting the token subsequence $\hat{x}_{b:e}$ therefore satisfies:

$$q_b(\hat{x}_b) \le q_{b+1}(\hat{x}_{b+1}) \le \cdots \le q_e(\hat{x}_e),$$
$$\text{where} \quad q_{e+1}(\hat{x}_{e+1}) < q_e(\hat{x}_e). \tag{2}$$

By iteratively applying this criterion across the entire draft sequence, we decompose it into multiple variable-length token subsequences, each representing a coherent local semantic unit. These subsequences then form the foundation for the subsequent semantic consistency verification step.

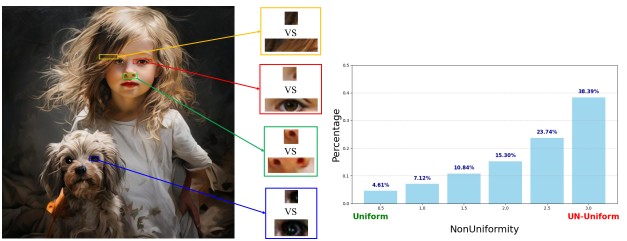

*(a)* Visualization analysis on sin-*(b)* Statistical analysis on re-gle token and token sequence. jected token subsequences.

*Figure 2.* Analysis of semantic verification on token subsequence.

## 4.2. Semantic Verification on Token Subsequence

Based on the above set of token subsequence achieved in Section 4.1, instead of performing token-by-token verification as existing SJD methods, we turn to perform verification on semantic-aware token subsequence level. The core challenge of achieving the above idea is how calculate the joint probability of token subsequence. A straightforward method is directly regarding the product of all token probability as the joint probability of token subsequence. That is,

$$q_{\mathcal{S}_k}(\hat{x}_{b:e}) = \prod_{i=b}^{e} q_{(}\hat{x}_i). \tag{3}$$

where $q_{\mathcal{S}_k}$ is the joint probability of $k$-th token subsequence $\mathcal{S}_k$, $b$ and $e$ are its start and end index on original sequence. Although the strategy is effective, in practice we find that it does not yield significant performance improvements. To analyze this reason, we conduct a statistical analysis on rejected token subsequences, as shown in Figure 2(b). We observe that most rejected subsequences are highly imbalanced in token probabilities: early tokens in the subsequence exhibit high uncertainty (indicating that they contribute little to the overall semantic meaning of the subsequence), whereas later tokens become highly certain (indicating that they dominate the semantic content of the subsequence). This suggests that tokens at different positions contribute unevenly to the semantic representation of the subsequence.

To mitigate the influence of highly uncertain early tokens, we propose an adaptive joint probability estimation strategy. Instead of treating all tokens equally, we identify the subset of tokens that collectively carry the majority of the subsequence's semantic information. Specifically, we adaptively truncate the potentially ambiguous prefix and retain the high-confidence suffix $\hat{x}_{bg:e}$. The starting position $bg$ is determined by a probability proportion threshold $\alpha \in (0, 1)$: we select the largest index $bg$ such that the cumulative confidence of the suffix $\hat{x}_{bg:e}$ accounts for at least $\alpha$ of the total probability of the entire subsequence $\hat{x}_{b:e}$:

$$\frac{\sum i = bg^{e} q_i(\hat{x}_i)}{\sum_{i=b}^{e} q_i(\hat{x}_i)} \geq \alpha. \tag{4}$$

---

**Algorithm 2** FallbackVerify

**Input:** Subsequence $\hat{x}_{b:e}$, Threshold $\alpha$
**Output:** Valid end index $e'$ (or $b - 1$ if failed)
1: Find largest $bg \in [b, e]$ such that $\frac{\sum_{i=bg}^{e} q_i(\hat{x}_i)}{\sum_{i=b}^{e} q_i(\hat{x}_i)} \geq \alpha$
2: $e' \leftarrow e$
3: **while** $e' \geq bg$ **do**
4:   $r \leftarrow \min\left(1, \frac{\prod_{i=bg}^{e'} q_i(\hat{x}_i)}{\prod_{i=bg}^{e'} p_i(\hat{x}_i)}\right)$
5:   **if** random $u \leq r$ **then**
6:     **return** $e'$   *// Accepted at current length*
7:   **end if**
8:   $e' \leftarrow e' - 1$   *// Fallback: shorten suffix*
9: **end while**
10: **return** $b - 1$   *// Verification failed*

---

This truncation effectively filters out tokens with dispersed probability distributions, preserving only the decisive semantic anchors of the subsequence. Finally, the joint probability of token subsequence can be calculated on the high-confidence suffix $\hat{x}_{bg:e}$. That is,

$$q_{\mathcal{S}_k}(\hat{x}_{b:e}) = \prod_{i=bg}^{e} q_{(}\hat{x}_i). \tag{5}$$

To further improve the acceptance rate, we incorporate a robust **Fallback** mechanism. As shown in Alg. 2, if the verification fails for the current high-confidence suffix, we do not immediately discard the entire sequence. Instead, we progressively remove the last token and iteratively re-verify the remaining prefix until a valid subsequence $\hat{x}_{bg:e'}$ (where $bg \leq e' < e$) is accepted or the sequence becomes empty.

Once the longest valid prefix $\hat{x}_{bg:e'}$ is determined, the token at the subsequent position $e' + 1$ is deemed invalid. We explicitly reject all drafted tokens after $e'$ and resample a token for position $e' + 1$ using the standard rectified distribution:

$$x_{e'+1} \sim \text{norm}(\max(0, q_{e'+1} - p_{e'+1})) \tag{6}$$

where $q$ and $p$ represent the target and draft probabilities, respectively. Other operation is consistent with SJD.

## 4.3. Theoretical Analysis of SJD-SV Effectiveness

The primary objective of SJD-SV is to enhance the theoretical acceptance probability of the drafted token sequence during the verification step. Next, we explain why our SJD-SV can achieve this goal from a theoretical perspective. Formally, let $q(\hat{x})$ and $p(\hat{x})$ denote the probabilities of the token $x$ in verification and draft step, respectively. A drafted token $\hat{x}_i$ is accepted if and only if a random variable $r \sim U[0, 1]$ satisfies the condition $r \leq min(1, \frac{q(\hat{x}_i)}{p(\hat{x}_i)})$.

*Table 1.* Results of quantitative evaluation on the Parti-prompt and MS-COCO 2017 benchmarks. The best results are highlighted in bold.

| Configuration | Latency (↓) | NFE (↓) | Acceleration (↑) | | FID (↓) | CLIP-Score (↑) |
|---|---|---|---|---|---|---|
| | | | **Latency** | **NFE** | | |
| **Parti-prompt** | | | | | | |
| Lumina-mGPT (Liu et al., 2024a) | 79.37s | 2392 | 1.00× | 1.00× | – | 32.09 |
| Jacobi Decoding (Song et al., 2021) | 82.21s | 2300.0 | 0.97× | 1.04× | – | 32.09 |
| SJD (Teng et al., 2024a) | 36.07s | 1035.3 | 2.20× | 2.31× | – | 32.09 |
| SJD-SV | **30.48s** | **826.32** | **2.60×** | **2.89×** | – | **32.13** |
| GSD (So et al., 2025) | 33.36s | 898.97 | 2.38× | 2.66× | – | 32.11 |
| GSD-SV | **30.77s** | **721.47** | **2.58×** | **3.32×** | – | **32.12** |
| LANTERN(Jang et al., 2024b) | 31.40s | **636.66** | 2.52× | 3.76× | – | 32.10 |
| LANTERN-SV | **27.72s** | **552.93** | **2.86×** | **4.33×** | – | **32.11** |
| **MS-COCO 2017** | | | | | | |
| Lumina-mGPT (Liu et al., 2024a) | 86.55s | 2379 | 1.00× | 1.00× | 30.79 | 31.31 |
| Jacobi Decoding (Song et al., 2021) | 85.64s | 2312 | 1.01× | 1.03× | 30.78 | 31.31 |
| SJD (Teng et al., 2024a) | 40.10s | 1058.6 | 2.16× | 2.25× | 30.78 | 31.31 |
| SJD-SV | **32.92s** | **866.80** | **2.63×** | **2.74×** | **30.76** | **31.33** |
| GSD (So et al., 2025) | 34.12s | 925.89 | 2.54× | 2.56× | 31.50 | 31.33 |
| GSD-SV | **29.96s** | **745.88** | **2.89×** | **3.19×** | **31.47** | **31.34** |
| LANTERN(Jang et al., 2024b) | 33.80s | 655.37 | 2.56× | 3.63× | 31.72 | 31.32 |
| LANTERN-SV | **23.11s** | **568.30** | **3.75×** | **4.19×** | **31.70** | **31.34** |

**The Derivation of Acceptance Probability.** For a drafted token sequence $\mathbf{c} = \{\hat{x}_1, \ldots, \hat{x}_k\}$, the baseline SJD performs verification independently for each token. This implies sampling $k$ independent random variables $\{r_1, \ldots, r_k\}$ for the $k$ tokens. The sequence is accepted only if the condition $r_i \leq \min(1, \frac{q(\hat{x}_i)}{p(\hat{x}_i)})$ holds for all $i \in [1, k]$. Therefore, the joint acceptance probability of SJD is defined as the product of probabilities of these independent events:

$$P_{\text{SJD}} = \prod_{i=1}^{k} P\left(r_i \leq \min\left(1, \frac{q(\hat{x}_i)}{p(\hat{x}_i)}\right)\right). \quad (7)$$

According to the definition of the **Cumulative Distribution Function (CDF)** of the standard uniform distribution, the probability $P(r \leq \alpha)$ is equal to $\alpha$ for any $\alpha \in [0, 1]$. Thus, Eq. 7 simplifies to the explicit value:

$$P_{\text{SJD}} = \prod_{i=1}^{k} \min\left(1, \frac{q(\hat{x}_i)}{p(\hat{x}_i)}\right). \quad (8)$$

In contrast, SJD-SV verifies the sequence as a single unit using a single random variable $r \sim U[0, 1]$. The entire sequence is accepted if it satisfies the condition $r \leq \min(1, \prod_{i=1}^{k} \frac{q(\hat{x}_i)}{p(\hat{x}_i)})$. Similarly, by applying the CDF property, the acceptance probability of SJD-SV is derived as:

$$P_{\text{SV}} = \min\left(1, \prod_{i=1}^{k} \frac{q(\hat{x}_i)}{p(\hat{x}_i)}\right). \quad (9)$$

**Theoretical Proof on $P_{\text{SV}} \geq P_{\text{SJD}}$.** To formalize the efficiency improvement, we compare Eqs. 1 and 2. According to the mathematical inequality for non-negative terms

$\min(1, \prod a_i) \geq \prod \min(1, a_i)$, we strictly establish the relationship between the two probabilities:

$$\min\left(1, \prod_{i=1}^{k} \frac{q(\hat{x}_i)}{p(\hat{x}_i)}\right) \geq \prod_{i=1}^{k} \min\left(1, \frac{q(\hat{x}_i)}{p(\hat{x}_i)}\right). \quad (10)$$

This inequality validates that $P_{\text{SV}} \geq P_{\text{SJD}}$, ensuring that SJD-SV theoretically achieves a higher acceptance rate for any given draft sequence.

**Distributional Consistency.** Consider a token sequence $\hat{x} = \{\hat{x}_1, \ldots, \hat{x}_k\}$ generated by the draft step. Let $p(\hat{x}) = \prod_{i=1}^{k} p(\hat{x}_i|\hat{x}_{<i})$ and $q(\hat{x}) = \prod_{i=1}^{k} q(\hat{x}_i|\hat{x}_{<i})$ denote the joint probabilities of all tokens within the sequence under the draft and verification steps, respectively. Since the draft and verification steps are independent, the probability of sequence $\hat{x}$ being both drafted and accepted, denoted as $P_{\text{out}}(\hat{x})$, is formulated as the product of its proposal probability and the joint acceptance criterion:

$$P_{\text{out}}(\hat{x}) \propto p(\hat{x}) \cdot \min\left(1, \frac{q(\hat{x})}{p(\hat{x})}\right) = q(\hat{x}). \quad (11)$$

Eq. 11 confirms that the resultant distribution of SJD-SV is identical to the verification distribution. Thus, SJD-SV is guaranteed to maintain distributional consistency.

## 5. Experiments

### 5.1. Experiment Setting

**Dataset** Experiments are conducted on two datasets: **Parti-prompt** (Yu et al., 2022) (1,600 curated prompts) spans

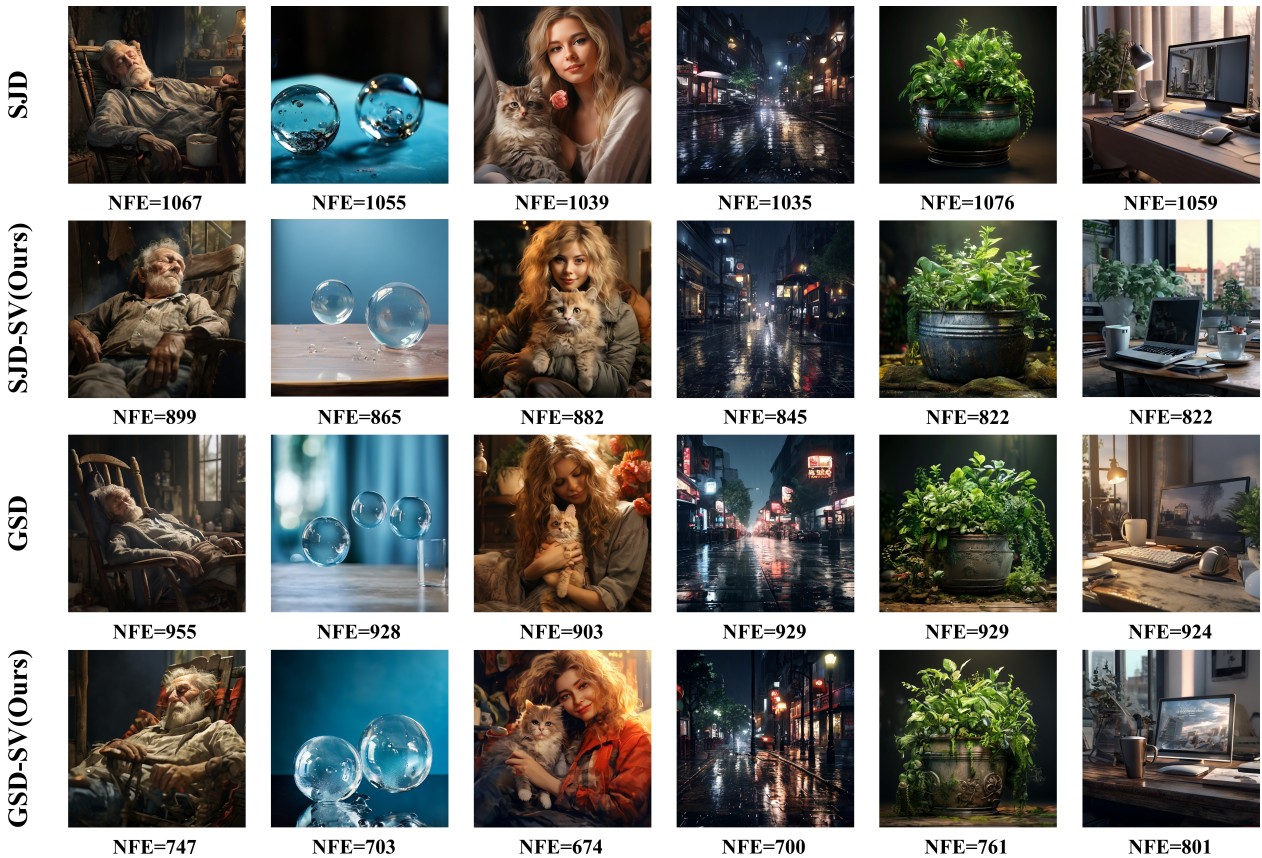

*Figure 3.* Qualitative experiment. Our method shows on average $1.25\times$ NFE acceleration while maintaining image quality.

diverse categories and complex compositions for evaluating artistic styles and challenging scenarios, while **MS-COCO 2017** (Lin et al., 2014) (118k image-caption pairs) provides natural language descriptions of scenes for assessing practical real-world generation capability.

**Baselines and Metrics** Five methods are compared: (1) **Lumina-mGPT** (Liu et al., 2024b), (2) **Jacobi Decoding** (Song et al., 2021), (3) **SJD** (Teng et al., 2024a), (4) **GSD** (So et al., 2025), and (5) **Lantern** (Jang et al., 2024a). Evaluation is performed from two key aspects. For generation efficiency, **NFE** (Number of Function Evaluations) (Song et al., 2020) and **Latency** are measured, with **Acceleration** ratios reported relative to Lumina-mGPT baseline (lower is better for NFE and Latency). For generation quality, **FID** (lower is better) (Heusel et al., 2017) measures overall visual fidelity and **CLIP-Score** (higher is better) measures semantic text-image alignment (Radford et al., 2021).

**Implementation Details** All experiments are conducted on a single NVIDIA A100 80GB GPU using Python with standard PyTorch and Huggingface Transformers. For our SJD-SV, we set the confidence score proportion threshold

*Table 2.* Ablation study on the token ambiguity issue.

| Method | Accept (All) | Accept (Amb.) | Accept (Clear) |
|---|---|---|---|
| SJD | 60.53% | 56.49% | 61.67% |
| **SJD-SV** | **64.16**% | **62.03**% | **65.83**% |

$\alpha = 0.8$ for all datasets. All other experimental settings, including backbone model, tokenizer, and hyperparameters, strictly follow GSD (So et al., 2025) for fair comparison.

### 5.2. Experimental Results

**Quantitative Evaluation** Table 1 presents quantitative results comparing baseline methods with their SV-enhanced variants on both benchmarks. The results show that integrating our proposed method consistently and significantly reduces generation latency and NFE across all baseline methods (including SJD, GSD, and LANTERN) while maintaining comparable image quality in terms of CLIP-Score and FID. The consistent improvements across diverse baseline architectures validate the plug-and-play nature of our method, demonstrating that it serves as a general acceleration module for autoregressive multimodal generation.

*Table 3.* Analysis of our fallback mechanism on Parti-prompt.

| Fallback | Latency | NFE | CLIP |
|---|---|---|---|
| w/o | 35.78s | 972.92 | 32.11 |
| **w/ (Ours)** | **30.48s** | **826.32** | **32.13** |

*Table 4.* Adaptive vs. fixed-length joint probability estimation.

| Strategy | Latency | NFE | CLIP |
|---|---|---|---|
| Fixed-1 | 26.29s | 710.83 | 31.04 |
| Fixed-2 | 28.84s | 760.92 | 31.76 |
| Fixed-3 | 30.53s | 822.11 | 32.09 |
| Fixed-4 | 32.74s | 867.37 | 32.11 |
| All | 33.25s | 877.29 | 32.08 |
| **Adaptive (Ours)** | **30.48s** | **826.32** | **32.13** |

*Table 5.* Ablation study of hyperparameter $\alpha$ on Parti-prompt.

| $\alpha$ | NFE | Latency | CLIP |
|---|---|---|---|
| 1.0 | 877.29 | 33.25s | 32.08 |
| 0.9 | 843.53 | 31.44s | 32.11 |
| **0.8** | **826.32** | **30.48s** | **32.13** |
| 0.7 | 790.84 | 29.13s | 32.06 |
| 0.6 | 748.26 | 27.38 | 32.01 |
| 0.5 | 670.80 | 24.68 | 31.88 |

*Table 6.* Computational overhead analysis on Parti-prompt.

| | SJD-SV | GSD-SV | LANTERN-SV |
|---|---|---|---|
| Time overhead | 2.05ms | 5.54ms | 0.81ms |

*Table 7.* Results of other baselines on the Parti-prompt benchmark. The best results are highlighted in bold.

| Configuration | Latency ($\downarrow$) | NFE ($\downarrow$) | Acceleration ($\uparrow$) Latency | Acceleration ($\uparrow$) NFE | FID ($\downarrow$) | CLIP-Score ($\uparrow$) |
|---|---|---|---|---|---|---|
| **Parti-prompt** | | | | | | |
| Emu3(Wang et al., 2024) | 780.98s | 8193.0 | 1.00× | 1.00× | – | 32.13 |
| Emu3(+SJD) | 345.65s | 3571.0 | 2.26× | 2.29× | – | 32.14 |
| Emu3(+Ours) | **180.30s** | **2786.1** | **4.33×** | **2.94×** | – | **32.16** |
| LLamagen(Sun et al., 2024a) | 30.58s | 927.5 | 1.00× | 1.00× | – | 28.14 |
| LLamagen(+SJD) | 19.86s | 569.0 | 1.54× | 1.63× | – | 28.16 |
| LLamagen(+Ours) | **19.43s** | **555.4** | **1.57×** | **1.67×** | – | **28.17** |

**Qualitative Evaluation** To evaluate generation quality across diverse scenarios, we designed text prompts covering six representative semantic dimensions: (1) **Human Structure** (anatomical correctness): *An old man sleeping in a rocking chair. 4k, realistic*; (2) **Object Counting** (numerical accuracy): *two light-blue bubbles on the table*; (3) **Subject Interaction** (action synthesis): a woman is taking a photo, holding a cute cat on her lap. 4k, very detailed, realistic; (4) **Outdoor Layout** (spatial composition): On a rainy night along the commercial street, the lights are faintly visible through the rain. 4k, realistic; (5) **Organic Texture** (natural patterns): *A pot of green plants, full of vitality. 4k, realistic, very detailed*; (6) **Indoor Layout** (spatial arrangement): *In the morning, a grey-shelled computer is in standby mode and placed on the table, and beside the computer on the table, there is a cup of coffee. 4k, realistic, very detailed.*

Figure 3 visualizes the qualitative results across these six semantic categories. Our methods maintain comparable visual quality to the baseline while achieving improved inference efficiency. The generated images preserve fine-grained details across all dimensions, demonstrating that our method effectively reduces computational cost without compromising generation fidelity.

## 5.3. Ablation Study

**Can our SJD-SV method effectively alleviate the token ambiguity issue?** To answer this question, in Table 2, we calculate the acceptance rate at different positions, i.e., with different semantic uncertainty. Specifically, we report the acceptance rate of all tokens, ambiguous tokens (i.e. early positions at the token subsequence), and the confident tokens (i.e. last positions at the token subsequence). From Table 2, we can see that while both methods achieve comparable acceptance rates at high-confidence positions, our method significantly and consistently improves the acceptance rate at ambiguous positions. This means that our method indeed can effectively mitigate the token confusion issue.

**Is our fallback mechanism effective?** To verify this point,

in Table 3, we evaluate the impact of our progressive fallback mechanism by comparing it against a baseline that rejects the entire sequence. From Table 3, we observe that while both methods achieve comparable generation quality, the baseline without fallback incurs significantly higher NFE. This is because discarding the entire sequence forces more frequent rejection and resampling, whereas our fallback mechanism remains the valid prefix and only resamples from the first invalid position, thereby reducing latency.

**Does adaptive joint probability estimation improve over fixed-length strategy?** We compare our adaptive joint probability estimation strategy against fixed-length strategies that verify only the last 1, 2, 3, 4 and all tokens. As shown in Table 4, fixed strategy (e.g., last-1) achieve relatively high efficiency but poor quality, while longer fixed suffixes improve quality at the cost of efficiency. Our adaptive method achieves the optimal balance, matching optimal quality as fixed-length strategy while maintaining superior efficiency.

**How does the hyperparameter $\alpha$ impact performance?** In Table 5, we analyze the impact of the hyperparameter $\alpha$ by setting it from $0.5$ to $1.0$. From Table 4, we can see that 1) comparing with $\alpha = 1.0$ (i.e., calculating the product of all token probability as joint probability), our method achieves significant efficiency improvement, which is because our method filters the influence of highly uncertain tokens; and 2) our SJD-SV achieves best on both image quality and efficiency when $\alpha = 0.8$, thus $\alpha = 0.8$ is set in all datasets.

**Does our method introduce significant computational overhead?** In Table 6, We calculate the introduced computa-

tional time by comparing the latency-per-NFE ratio between our method and existing baseline. From results, we can see that our method only introduce marginal time overhead (around $1 \sim 5$ ms), as they primarily involve lightweight subsequence partition, which is essentially negligible compared to the expensive forward of the target model.

**Does our SJD-SV method generalize to other baseline models?** In order to get a more comprehensive evaluation of the performance of our proposed method, we integrated our proposed method with the other two baselines, Emu3 (Wang et al., 2024) and LLamagen (Sun et al., 2024a), and then we conduct experiment and compared the performance with that of the baselines themselves as well as the baselines that incorporated the SJD mechanism. The experimental results are shown in Table 7. As shown, our method significantly improves inference acceleration over the standard SJD on both models. This demonstrates that SJD-SV serves as a highly adaptable, plug-and-play solution for accelerating diverse generative architectures.

## 6. Conclusions

In this paper, we figure out the root reason causing the **token ambiguity** issue of existing visual autoregressive model, i.e., single visual token often correspond to unclear local patches, such that it is insufficient to accurately and unambiguously express a specific semantic concept. To address this issue, we propose **SJD-SV**, a novel acceleration framework that shifts the verification strategy from the token level to the **semantic-aware subsequence level**. In particular, a simple yet effective semantic-aware subsequence partition, an adaptive joint probability estimation, and a robust fallback mechanism are designed to ensure generation stability. Experimental results verify the effectiveness of our SJD-SV serving as a highly effective plug-and-play module.

## Impact Statement

This paper presents work whose goal is to advance the field of Machine Learning. There are many potential societal consequences of our work, none which we feel must be specifically highlighted here.

## Acknowledgements

This work was supported by the National Nature Science Foundation of China under Grant No. 62502120, 62272130, 62376072, and 62302127, the Guangdong Basic and Applied Basic Research Foundation under Grant No. 2025A1515011674, Shenzhen Science and Technology Program No. QNXMC20250701091911015, ZDCYKCX202509001092700001, KCXFZ2024090309-3006009, KCXFZ20230731094905010, and SYSPG2024-

1211173609009.

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
