# OpenReview forum: "SJD-SV: Speculative Jacobi Decoding with Semantics Verification for Autoregressive Image Generation"
_ICML.cc/2026/Conference — ICML 2026 regular_

### Official Review · Reviewer_uTxB · 2026-03-07

**Soundness:** 3
**Presentation:** 3
**Significance:** 2
**Originality:** 3
**Overall Recommendation:** 4
**Confidence:** 3

**Summary:**

The paper addresses the high inference latency of AR image generation models. While Speculative Jacobi Decoding (SJD) accelerates this process, it often rejects valid tokens due to "token ambiguity," a phenomenon where fine-grained visual tokens lack strong individual semantic meaning because they represent very small local patches. To resolve this, the authors propose Speculative Jacobi Decoding with Semantics Verification (SJD-SV), which evaluates tokens as semantic-aware subsequences rather than individually. The method partitions tokens based on non-decreasing probability trends and employs an adaptive joint probability estimation paired with a fallback mechanism. Experiments on the Parti-prompt and MS-COCO 2017 datasets demonstrate that integrating SJD-SV into existing baselines improves acceleration (up to a ~4.33x NFE reduction) without degrading image quality.

**Compliance With Llm Reviewing Policy:**

Affirmed.

**Final Justification:**

The rebuttal solved my concerns.

**Key Questions For Authors:**

Please refer to the weaknesses.

**Limitations:**

No. The authors should claim the limitations of the work.

**Strengths And Weaknesses:**

Strengths:

- Insightful Problem Diagnosis: The authors provide a clear visualization and explanation for token ambiguity in visual AR models, correctly identifying that individual tokens represent small, semantically under-specified local image patches.

- Novel Verification Granularity: Shifting the verification step from individual tokens to semantic-aware token subsequences is a logical, well-executed, and highly effective solution.

- Theoretical Guarantees: The paper provides a solid theoretical foundation, mathematically proving that the acceptance probability of SJD-SV ($P_{SV}$) is strictly greater than or equal to standard SJD ($P_{SJD}$). They also effectively prove distributional consistency.

- Strong Empirical Results and Plug-and-Play Nature: The method is training-free and integrates seamlessly into existing SJD variants. It demonstrates consistent NFE and latency reductions across baselines like GSD and LANTERN while maintaining competitive FID and CLIP scores.

Weaknesses:

- Buried Key Results: While the method claims to be universally applicable, experiments on alternative AR models (Emu3 and LlamaGen) are relegated to the appendix. Given that generalizability is a core claim, at least a summary of these multi-model results should be present in the main text to strengthen the paper's narrative.

- Reliance on Heuristic Thresholds: The adaptive joint probability estimation relies heavily on a fixed probability proportion threshold ($\alpha=0.8$). While ablated, relying on a hard-coded threshold may limit generalization across entirely different architectures or highly

- Limited Computational Overhead Analysis: The time overhead analysis in Table 6 only reports minor latency differences in milliseconds on the Parti-prompt dataset. The manuscript lacks a deeper exploration of how this computational overhead scales with significantly longer sequence lengths or larger batch sizes.

---

> ### Author Rebuttal · Authors · 2026-03-29
>
> ### To Weakness#1
> Thank you for this helpful suggestion. In the final version, we will include a summary of the results for these alternative AR models (Emu3 and LlamaGen), currently shown in the Appendix, in the main text to better strengthen the paper’s narrative.
>
> ### To Weakness#2
> Thank you for this valuable suggestion. This insightful comment motivates us to further explore adaptive or more generalizable approaches for probability thresholds in future work.
>
> ### To Weakness#3
>
> Thank you for this valuable comment. Following your suggestion, we conducted an additional analysis with longer sequence lengths on the Parti-Prompt dataset. The results show that our method consistently improves performance across all tested sequence lengths, with the relative benefit becoming more pronounced as the sequence length increases. This further verifies the effectiveness of our method.
> |    Method     | Latency (↓) | NFE (↓) | Clip Score (↑) |
> | :-----------: | :---------: | :-----: | :------------: |
> |  SJD (L=16)   |  36.07s           |   1035.3      |    32.09            |
> | SJD-SV (L=16) |  30.48s           |   826.32      |    32.13            |
> |  SJD (L=32)   |  35.30s           |   1012.72      |    32.08            |
> | SJD-SV (L=32) |  29.31s           |   789.20      |    32.13            |
> |  SJD (L=64)   |  35.93s           |   1025.4      |     32.08           |
> | SJD-SV (L=64) |  29.19s           |   783.48      |     32.11           |

---

> > ### Author Rebuttal · Reviewer_uTxB · 2026-04-02
> >
> > I am wondering why the latency is lower when having longer prompts.

---

> > > ### Author Response · Authors · 2026-04-03
> > >
> > > Thank you for the insightful question. We would first like to sincerely apologize for the confusion in our previous response. In the above Table, the variable *L* does **not** represent the length of the input prompt, but rather the **draft window size** in SJD-SV. We regret that we misunderstood the reviewer’s concern and greatly appreciate the opportunity to clarify this point.
> > >
> > > **Why does a larger draft window lead to lower latency?**
> > > The observed latency reduction with increasing *L* stems from improved efficiency in the draft-based generation process. Specifically, a larger draft window enables more tokens to be proposed and verified in each iteration, thereby reducing the total number of autoregressive decoding steps. Although each iteration becomes slightly more computationally intensive, this cost is amortized over fewer steps, resulting in a net reduction in latency. This is also consistent with the decrease in NFE shown in the table, which directly reflects fewer model evaluations.
> > >
> > > **Impact of prompt length (new experiment).**
> > > To properly address the reviewer’s original concern, we conducted an additional experiment to evaluate the effect of **prompt length** on performance. Considering that text prompt lengths in the Parti-Prompt dataset range from 1 to 67 words, we partitioned them into three groups: `<20`, `20–40`, and `>40` words and compared SJD-SV and SJD under each setting. The results are summarized below:
> > >
> > > | Method        | Text Prompt Length | NFE    | Latency |
> > > |---------------|--------------|--------|---------|
> > > | Lumina-mGPT (token-by-token)   | <20          |  2392   | 79.06s  |
> > > | Lumina-mGPT (token-by-token)  | 20–40       | 2392   | 79.38s  |
> > > | Lumina-mGPT (token-by-token)  | >40         | 2392   | 79.41s  |
> > > | SJD           | <20          | 1029.1 | 35.73s  |
> > > | SJD           | 20–40       | 1035.2 | 36.05s  |
> > > | SJD           | >40         | 1042.5 | 36.39s  |
> > > | SJD-SV        | <20          | 820.71 | 30.21s  |
> > > | SJD-SV        | 20–40       |  826.35 | 30.50s  |
> > > | SJD-SV        | >40         | 833.15 | 30.92s  |
> > >
> > > As shown above, **prompt length has small impact on both NFE and latency**, with only a very minor increase in runtime as the prompt becomes longer. This behavior can be attributed to the Transformer-based autoregressive architecture. While longer prompts increase the sequence length withinach forward pass, these computations are highly parallelized, and the additional cost per step is relatively small compared to the overall generation cost dominated by the number of decoding steps. Therefore, the total latency remains largely unaffected.
> > >
> > > Importantly, SJD-SV consistently achieves stable acceleration over SJD across all prompt length ranges, further demonstrating the robustness and generality of our method.

---

### Official Review · Reviewer_f226 · 2026-03-10

**Soundness:** 1
**Presentation:** 2
**Significance:** 2
**Originality:** 3
**Overall Recommendation:** 4
**Confidence:** 3

**Summary:**

This paper proposes SJD-SV, a plug-in module for accelerating autoregressive image generation based on the Speculative Jacobi Decoding (SJD) framework. Specifically, SJD-SV performs token verification on a semantic-aware subsequence (i.e., the drafted token sequence). Furthermore, the method introduces a Fallback mechanism that progressively removes the trailing token and iteratively re-verifies the remaining prefix until a valid subsequence is obtained.

Extensive experimental results demonstrate that SJD-SV achieves significant performance improvements on base SJD, and theoretical analysis further guarantees the acceptance rate and distributional consistency of the proposal.

**Compliance With Llm Reviewing Policy:**

Affirmed.

**Final Justification:**

The authors have adequately addressed my concerns in the rebuttal.

**Key Questions For Authors:**

(1)	The paper's claim that “token subsequences identified using this probability-increasing feature correspond to more semantically coherent units compared to individual tokens” is supported only by a visualization of a few tokens from a single image (Figure 2), which is insufficient to establish a general conclusion. A deeper explanation would resolve this question.

(2)	There are several studies proposed to optimize SJD for improving token acceptance rates in image generation. The authors could compare with these methods, such as SJD++ [1] and MC-SJD [2], to further demonstrate its effectiveness.

(3)	In Sec. 5.3, a higher acceptance rate alone does not directly demonstrate that the proposed method effectively mitigates the token confusion issue, visualization of the accepted tokens may better demonstrate this conclusion.

(4)	The authors should carefully check and correct the grammatical and notation errors in the manuscript.

**Limitations:**

yes

**Strengths And Weaknesses:**

**Strengths:**

(1)	This paper analyzes the token ambiguity problem through visualization and proposes performing verification at the semantic-aware token subsequence level to accelerate image generation, thereby improving the token acceptance rate.

(2)	The proposed method is simple and plug-in, accelerating image generation without introducing significant additional overhead.

(3)	Experiments demonstrate that the proposed method effectively improves the image generation efficiency of SJD, and the theoretical analysis further proves its effectiveness.

**Weaknesses:**

(1)	The paper's claim that ''token subsequences identified using this probability-increasing feature correspond to more semantically coherent units compared to individual tokens'' is supported only by a visualization of a few tokens from a single image (Figure 2), which is insufficient to establish a general conclusion.

(2)	There are several studies proposed to optimize SJD for improving token acceptance rates in image generation. However, the authors only compare against the base SJD and do not compare with other optimization methods, such as SJD++ [1] and MC-SJD [2].

(3)	In Sec. 5.3, a higher acceptance rate alone does not directly demonstrate that the proposed method effectively mitigates the token confusion issue, as there is no intuitive causal link between them (an increased acceptance rate may simply result from relaxed verification criteria).

(4)	Poor writing. There are several grammatical errors and inconsistencies in notation across sections. Such as, correction -> correlation (Abstract), represents -> present (Figure 1), individual tokens is -> individual tokens are (Figure 1), $x$ denotes a conditioning input and $y$ denotes the token sequence in Sec. 3.1, while $x$ denotes token in Sec. 4.

[1] Teng Y, Jiang Z, Shi H, et al. SJD++: Improved Speculative Jacobi Decoding for Training-free Acceleration of Discrete Auto-regressive Text-to-Image Generation[J]. arXiv preprint arXiv:2512.07503, 2025.

[2] So J, Kook H, Jang C, et al. MC-SJD: Maximal Coupling Speculative Jacobi Decoding for Autoregressive Visual Generation Acceleration[J]. arXiv preprint arXiv:2510.24211, 2025.

---

> ### Author Rebuttal · Authors · 2026-03-29
>
> ### To Question#1
> We thank the reviewer for the constructive comment. Our claim is based on the underlying idea that, in autoregressive generation, a subsequence whose probability increases monotonically reflects accumulating model confidence and strong semantic dependency among the tokens. This principle suggests that such subsequences are more likely to form coherent semantic units than individual tokens. We illustrate this phenomenon using a visual example in Figure 2(a). From Figure 2, we observe that, compared to individual tokens, subsequences tend to correspond to image regions with clearer and more coherent semantic structures. This is because each subsequence aggregates multiple correlated tokens (i.e., non-decreasing probability trends), which together provide a more complete and expressive representation of the underlying visual semantics, rather than relying on isolated token. While we have considered more explicit or quantitative ways to demonstrate this effect, the extremely fine-grained nature of visual tokens makes it challenging to directly quantify their semantic content in a clear and interpretable manner.
>
> To provide additional intuition from a more interpretable domain, this effect can also be understood through an analogy in natural language. In this analogy, token subsequences in our method are similar to common phrases in NLP, while individual tokens correspond to words. For example, when generating the phrase “United States of America,” once the model has produced the tokens “United States of,” the next token “America” becomes highly predictable and is assigned a substantially higher probability than other candidates. It is well known that phrases typically convey clearer and more coherent semantics than isolated words. By analogy, subsequences with increasing probabilities are more likely to capture coherent semantic structures than individual tokens, providing intuitive support for our claim. We will clarify this point in the final version.
>
> ### To Question#2
> Referring to the valuable suggestion, we have integrated our plug-in SJD-SV method with other SJD optimization methods on the Parti-Prompt dataset, including SJD++ [1] and MC-SJD [2]. The results show that applying our method further improves the decoding efficiency of both SJD++ [1] and MC-SJD [2] without sacrificing image quality.
>
> |    Method     | Latency(↓) | NFE(↓) | Clip(↑) |
> | :-----------: | :--------: | :----: | :-----: |
> |     SJD++     |   27.12s         |  718.21      |  32.08       |
> | **SJD++ (+ ours)**  |   25.14s         |  625.11      |    32.09     |
> |    MC-SJD     |    29.83s        |  790.16      |    32.11     |
> | **MC-SJD (+ ours)** |    26.03s        |  675.22      |    32.12     |
>
> ### To Question#3
>
> Thank you for this insightful comment. A higher acceptance rate alone is not sufficient to directly demonstrate mitigation of token confusion. As discussed in the paper, semantic/token confusion primarily arises from a dispersed token probability distribution, where the probability mass is spread across multiple competing candidates. To provide more direct evidence, we conduct an additional experiment on the Parti-Prompt dataset by reporting the entropy of the probability distribution of accepted tokens.
>
> From the table, we observe that the entropy of the token distribution is significantly reduced when using our SJD-SV method, indicating a sharper and more concentrated distribution. This result suggests that our method effectively alleviates semantic/token confusion. We will update it to the final version.
>
> |   Method   | Average Entropy | **Relative Reduction (%)** |
> | :--------: | :-----: | :------------------------: |
> |    SJD     |  5.94       |             -              |
> | **SJD-SV** |  4.58       |        22.90%                    |
>
> ### To Question#4
>
> Thank you for pointing these writing issues. We will carefully proofread the manuscript and correct these issues in the final version.

---

> > ### Author Rebuttal · Reviewer_f226 · 2026-04-03
> >
> > The authors have adequately addressed my concerns in the rebuttal.
> > I appreciate the additional clarifications, and I will raise my rating accordingly.

---

### Official Review · Reviewer_oGN8 · 2026-03-13

**Soundness:** 3
**Presentation:** 3
**Significance:** 3
**Originality:** 3
**Overall Recommendation:** 4
**Confidence:** 3

**Summary:**

SJD-SV accelerates autoregressive image generation by shifting speculative verification from individual tokens to semantic-aware subsequences, effectively reducing token ambiguity and false rejections. It acts as a training-free plug-in that enhances existing SJD methods. Overall, the authors examine a central aspect of autoregressive acceleration, and this research aims to focus on a central area of generative inference efficiency.

**Compliance With Llm Reviewing Policy:**

Affirmed.

**Key Questions For Authors:**

1. Could you provide a broader statistical analysis across the dataset validating that token probabilities continuously increase within a semantic unit?
2. In Equation 4, is the search strategy for the index $bg$ performed linearly from the start or backwards from the end?
3. How does the relative capacity difference between the draft and target models affect the SJD-SV acceptance rate compared to standard SJD?
4. Is the random variable $u$ in Algorithm 2 a standard rejection sampling step or specifically tailored for the subsequence fallback logic?

**Limitations:**

Yes

**Strengths And Weaknesses:**

Strengths:
- Sound mathematical proof shows the acceptance probability of subsequence verification is theoretically higher than single-token verification.
- The method is fully training-free and serves as a plug-and-play module for existing speculative decoding frameworks, which means it's easy to be applied to real-world deployment scenarios.
- Extensive empirical results validate robust NFE and latency reductions without compromising generation quality.

Weaknesses:
- The assumption that token probabilities strictly increase within every semantic unit may fail during highly stochastic generation steps.
- The absolute incremental acceleration on top of the best baselines is solid but evolutionary rather than revolutionary.
- The core conclusion that token ambiguity purely stems from fine-grained patches may oversimplify complex factors like model learning capacity.

---

> ### Author Rebuttal · Authors · 2026-03-29
>
> ### To Weakness#1
> Thank you for this insightful comment. The assumption of strictly increasing token probabilities within a semantic unit can indeed be violated under highly stochastic generation, and such violations may become more likely as the subsequence length increases. It is this point that motivates us to design the fallback mechanism (see Section 4.2 for details). When a subsequence fails the validation criterion, the method does not discard it entirely; rather, it iteratively removes suffix tokens to shorten the subsequence and re-validates, thereby enabling adaptive recovery of valid segments. This mechanism allows the approach to remain robust to stochastic fluctuations in token probabilities. To verify its effectiveness, we conduct an ablation study in Table 3, where removing the fallback mechanism leads to consistent performance degradation. We will further clarify this point in the final revision.
>
> ### To Weakness#2
> Thank you for this valuable comment, which motivates us to conduct a deeper exploration of semantic verification in future work. We would also like to emphasize that the key idea of our method is to introduce a new perspective on subsequence-level validation and adaptive fallback, which is orthogonal to existing approaches and can be readily integrated with them. We will further clarify this point in the final revision.
>
>
> ### To Weakness#3
> Thank you for this constructive comment, which provides valuable insight. While our work highlights fine-grained patches as a source of token ambiguity, model capacity may also play an important role. This observation motivates future work exploring how improvements in model capacity could further accelerate image generation. We will clarify this perspective in the final revision.
>
> ### To Question#1
> Referring to this suggestion, we conduct a broader statistical analysis of token sequences generated in the standard token-by-token manner. Specifically, we analyze token sequences generated by the lumina-mgpt autoregressive model (i.e., token-by-token autoregressive generation) over a dataset of 1,000 images, comprising a total of over 1 million tokens. As shown in results, approximately 83% of subsequences exhibit locally non-decreasing probability trends, while only 16% correspond to decreasing subsequences. This indicates that non-decreasing probability patterns are highly prevalent across datasets (Parti-prompt and MS-COCO) and not limited to isolated examples. To further illustrate the visual semantics, in Figure 2, we provides an example visualization of these subsequences. From Figure 2, we observe that, compared to individual tokens, subsequences tend to correspond to image regions with clearer and more coherent semantic structures. This is because each subsequence aggregates multiple correlated tokens (i.e., non-decreasing probability trends), which together provide a more complete and expressive representation of the underlying visual semantics, rather than relying on isolated token.
>
> ​        **The proportion of non-decreasing vs. decreasing subsequences.**
>
> |     Datasets      | Non-decreasing Subsequence | Decreasing Subsequence |
> | :------------: | :----------------------------------: | :------------------------------: |
> | **Parti-prompt** |                83.79%                |              16.21%              |
> | **MS-COCO** |                83.01%                |              16.99%              |
>
> ### To Question#2
> In Eq. 4, the search for the index \(b_g\) is performed **backwards from the end** of the subsequence.
>
> ### To Question#3
> We guess that there may be a misunderstanding regarding the draft and target models in SJD-SV. In fact, the draft and target models are the same, so there is no relative capacity difference between them. We will make this distinction clearer in the revised version.
>
> ### To Question#4
>
> The random variable \(u\) in Algorithm 2 is part of the **standard rejection sampling procedure**, rather than a component specifically designed for the subsequence fallback logic. We will clarify this point in the final version.

---

> > ### Author Rebuttal · Reviewer_oGN8 · 2026-04-02
> >
> > My concerns have been resolved, and I will maintain the rating.

---

### Official Review · Reviewer_zvwi · 2026-03-16

**Soundness:** 2
**Presentation:** 3
**Significance:** 2
**Originality:** 2
**Overall Recommendation:** 4
**Confidence:** 3

**Summary:**

This paper proposes SJD-SV, a semantic verification framework which moves verification from the individual-token level to the token-subsequence level. Concretely, the method partitions drafted tokens into variable-length subsequences using a non-decreasing token-probability trend, then performs subsequence-level verification with an adaptive joint probability estimation and a fallback mechanism when verification fails.

**Compliance With Llm Reviewing Policy:**

Affirmed.

**Final Justification:**

The rebuttal solves my concern.

**Key Questions For Authors:**

Please refer to the weakness I list.

**Limitations:**

No.

**Strengths And Weaknesses:**

**Strength**

1. The underlying motivation is intuitive.

2. A strong practical advantage is that SJD-SV is presented as a plug-in module for existing SJD-style methods.

**Weakness**

1. The “semantic-aware” claim feels stronger than what the method actually implements. The subsequence partition is based on a non-decreasing probability trend of drafted tokens, where the evidence is fairly qualitative which is not solid enough.

2. Section 4.3 argues that subsequence-level acceptance probability is higher than token-wise verification under the proposed formulation, but the practical algorithm also includes adaptive truncation of the subsequence and a fallback mechanism that shortens the suffix until acceptance. Those practical design choices are important to the final method, yet the proof sketch does not seem to fully establish guarantees for the complete algorithmic pipeline.

3. The related work on autoregressive generation acceleration is somewhat incomplete. It does not sufficiently cover a broader line of work on parallel autoregressive generation. In particular, relevant directions such as PAR, LPD, NAR, RandAR, and ARPG are not discussed.

---

> ### Author Rebuttal · Authors · 2026-03-29
>
> ### To Weakness#1
>
> The goal of claiming “semantic-aware” is to emphasize that we perform the verification step on subsequences with non-decreasing probability rather than on individual tokens, as such subsequences tend to exhibit clearer visual semantics. This design is motivated by the intuition that a non-decreasing probability trend reflects increasing model confidence during generation, suggesting stronger dependency among consecutive tokens. As a result, the grouped tokens (i.e., subsequences) are more likely to form coherent semantic units compared to independent tokens, which helps alleviate token-level semantic ambiguity.
>
>  **The proportion of non-decreasing vs. decreasing subsequences.**
>
> |     Dataset      | Non-decreasing Subsequence | Decreasing Subsequence |
> | :------------: | :----------------------------------: | :------------------------------: |
> | **Parti-prompt** |                83.79%                |              16.21%              |
> | **MS-COCO** |                83.01%                |              16.99%              |
>
> To provide more quantitative support for this intuition, we further analyze token sequences generated by a recent autoregressive model (i.e., token-by-token autoregressive generation) over 1,000 images. We observe that a large proportion (around 83%) of sequences exhibit locally non-decreasing probability trends, indicating that such patterns are prevalent in practice. In addition, in Figure 2, we provides a visualization example for these subsequences. From Figure 2, we observe that, compared to individual tokens, subsequences correspond to image regions with clearer semantic structures.
> For these reasons, we use the term “semantic-aware” to describe our method. We will clarify this point in the final revision.
>
>
>
>
> ### To Weakness #2
>
> The adaptive truncation (fallback) mechanism is included to ensure robustness: if a subsequence fails verification, the suffix is shortened until acceptance. Importantly, this mechanism only reduces the length of the subsequence, with the worst-case scenario being single-token verification. Therefore, to demonstrate that our method increases acceptance probability, it suffices to show that verification on subsequences of length ≥ 2 has higher acceptance probability than single-token verification, which is precisely what our proof sketch establishes.
>
> To further clarify the practical advantage of our approach, we analyze the lengths of subsequences that are ultimately accepted after applying the fallback mechanism. We observe that nearly 71.43% of verified subsequences have length greater than 1:
>
>  **The proportion of accepted subsequence with different length L.**
> | Accepted subsequence length L | Percentage |
> | :---------------------------: | :--------: |
> |            L =  1             |  28.57%          |
> |             L > 1             |  71.43%          |
>
> This indicates that (1) the fallback mechanism rarely reduces subsequences to single tokens, and (2) compared to token-by-token verification, our method achieves the acceleration benefit on nearly 71.43% of subsequences. This further confirm the effectiveness of our approach.
>
>
>
> ### To Weakness #3
>
> We thank the reviewer for pointing out these relevant works. We would like to clarify that methods such as PAR, LPD, NAR, RandAR, and ARPG primarily focus on modifying the autoregressive generation process, which requires retraining the model. These methods thus fall under the “model-based acceleration” category that we already discuss in our related work. In contrast, our work focuses on training-free acceleration methods, which do not modify the autoregressive model itself and can be directly applied to existing models for inference acceleration. We will update the related work section in the final version.

---

> > ### Author Rebuttal · Reviewer_zvwi · 2026-04-03
> >
> > Thank you for the rebuttal. I will raise my score.

---

### Decision · Program_Chairs · 2026-04-30

**Decision:**

Accept (regular)

**Comment:**

The reviewers overall agree that this paper makes a solid contribution to accelerating autoregressive image generation. There were some minor concerns, but the rebuttal addressed these points well. After the rebuttal, the reviewers' consensus is clearly positive. Therefore, I recommend acceptance.